# Deep Learning Capabilities for the Categorization of Microcalcification

**DOI:** 10.3390/ijerph19042159

**Published:** 2022-02-14

**Authors:** Koushlendra Kumar Singh, Suraj Kumar, Marios Antonakakis, Konstantina Moirogiorgou, Anirudh Deep, Kanchan Lata Kashyap, Manish Kumar Bajpai, Michalis Zervakis

**Affiliations:** 1Machine Vision and Intelligence Lab, Department of Computer Science and Engineering, National Institute of Technology, Jamshedpur 831014, India; koushlendra.cse@nitjsr.ac.in (K.K.S.); sksuraj2136@gmail.com (S.K.); anirudhdeep.11@gmail.com (A.D.); 2Digital Image and Signal Processing Laboratory, School of Electrical and Computer Engineering, Technical University of Crete, 73100 Crete, Greece; dina@display.tuc.gr (K.M.); michalis@display.tuc.gr (M.Z.); 3Department of Computer Science and Engineering, Vellore Institute of Technology University, Bhopal 466114, India; kanchan.k@vitbhopal.ac.in; 4Computer Science and Engineering Discipline, PDPM Indian Institute of Information Technology Design Manufacturing, Jabalpur 482005, India; mkbajpai@iiitdmj.ac.in

**Keywords:** cancer, microcalcification, convolution neural network, biomedical imaging, mammograms

## Abstract

Breast cancer is the most common cancer in women worldwide. It is the most frequently diagnosed cancer among women in 140 countries out of 184 reporting countries. Lesions of breast cancer are abnormal areas in the breast tissues. Various types of breast cancer lesions include (1) microcalcifications, (2) masses, (3) architectural distortion, and (4) bilateral asymmetry. Microcalcification can be classified as benign, malignant, and benign without a callback. In the present manuscript, we propose an automatic pipeline for the detection of various categories of microcalcification. We performed deep learning using convolution neural networks (CNNs) for the automatic detection and classification of all three categories of microcalcification. CNN was applied using four different optimizers (ADAM, ADAGrad, ADADelta, and RMSProp). The input images of a size of 299 × 299 × 3, with fully connected RELU and SoftMax output activation functions, were utilized in this study. The feature map was obtained using the pretrained InceptionResNetV2 model. The performance evaluation of our classification scheme was tested on a curated breast imaging subset of the DDSM mammogram dataset (CBIS–DDSM), and the results were expressed in terms of sensitivity, specificity, accuracy, and area under the curve (AUC). Our proposed classification scheme outperforms the ability of previously used deep learning approaches and classical machine learning schemes.

## 1. Introduction

Cancer is a disease caused by an uncontrollable growth of abnormal cells without destroying the older and damaged cells. Cancer cells grow and divide in an uncontrolled manner, invading normal tissues and organs and eventually spreading throughout the body [1]. Breast cancer is categorized into (1) noninvasive/in situ and (2) invasive/infiltrating. Noninvasive breast cancer remains in the particular location of the breast without spreading to surrounding tissues, lobules, or ducts. Cancerous cells spread throughout the body using the blood or lymphatic systems, destroying healthy tissue in the process called invasion. Noninvasive breast cancer is classified as ductal and lobular. Ductal carcinomas emanate from ducts, whereas lobular carcinomas emanate from lobules, and it does not eradicate other tissues [2]. Ductal carcinoma in situ (DCIS) is the most general form of noninvasive carcinoma. It is called noninvasive because it does not disseminate apart from the milk duct into surrounding normal breast tissues. Microcalcification is the most common mammographic indication of DCIS. DCIS is not severe, but it can lead to invasive breast cancer after some time [1,3]. Microcalcifications are small size deposits of calcium, which is brighter, compared with normal breast tissues. The size of microcalcification ranges from 100 microns up to 2000 microns [4]. They might appear isolated or in form of a cluster [4]. Isolated microcalcifications are difficult to detect due to their small size and low contrast. Microcalcification clusters make patterns and are detectable in the early screening of breast cancer. Microcalcification may be benign or malignant according to size, density, form, and distribution. Benign microcalcifications are of round shape, distributed in a diffused manner, and scattered in breast tissues. Malignant microcalcifications are distributed in an arbitrary pattern, irregular shape, and variable density. For these reasons, even highly experienced medical doctors need to carefully read and evaluate the mammograms, and therefore, automated systems effectively trained to trigger the suspicious areas become essential in clinical practice. In this paper, deep learning is used for automatic detection and classification of three categories of microcalcification—namely, (1) benign, (2) malignant, and (3) benign without callback. To reduce the training time of the deep learning model, the region of interest (ROI) that consists of microcalcification is manually cropped. Existing techniques of microcalcification detection exploit feature extraction and machine learning techniques. In several studies, authors have applied various image processing and machine learning techniques for the segmentation and detection of microcalcification. Heinlein et al. proposed integrated wavelet transform for microcalcification enhancement in mammograms [2]. Multiresolution decomposition-based approaches have earlier been applied to mammograms enhancement. Mini et al. applied wavelet-based techniques for microcalcification detection and achieved 95% accuracy on the MIAS dataset [3]. Bocchi et al. used Radon transform-based features for shape analysis of microcalcification to detect its malignancy by using Radon and moment-based features [4]. Papadopoulos et al. used contrast limited adaptive histogram equalization (CLAHE), local range modification, 2D redundant dyadic wavelet transform using linear stretching, and wavelet shrinkage technique for microcalcification enhancement [5]. Sakka et al. applied a multi-resolution-based wavelet technique for the detection of microcalcification on the MIAS dataset [6]. The Hessian matrix within a multi-resolution-based approach has been applied by Balakumarana et al. for microcalcification detection [7]. A high 98.3% true positive ratio with a 0.9% false-positive ratio is achieved on 100 mammograms of the DDSM dataset. Liu et al. achieved 92% sensitivity by applying probability-based fuzzy learning and weighted support vector machines (SVMs) for microcalcification detection on digital mammograms [8].

Recently, deep learning approaches with/or without transfer learning have also been used by various authors for object detection, microcalcification, and mass detection. Transfer learning exploits the knowledge obtained from other domains in the form of pretrained feature extraction layers, in order to overcome the need for large-size datasets in training. In this form, the network uses the available data for either fine-tuning its training parameters or the outmost classifier layer on the domain of interest. Wang et al. have applied a context-sensitive-based deep neural network for microcalcification detection [9]. Jiao et al. applied intensity information and deep features extracted by using a deep convolution neural network (CNN) for mass classification [10]. Ribli et al. used a faster R-CNN deep learning model to detect and classify breast lesions using mammograms. The highest AUC value of 0.95 has been achieved on IN Breast dataset [11]. Arevalo et al. used a CNN-based approach for mass classification, achieving the highest AUC of 0.82 on 736 film mammograms [12]. Dhungel et al. applied a deep-learning-based model to analyze the masses present in the mammograms, and 98% sensitivity was achieved [13]. Becker et al. investigated a deep artificial neural network model for breast cancer detection on digital mammograms and achieved 81% accuracy at its highest [14].

### Focus and Contribution of the Present Study

In this study, we investigated the potential of deep learning in the automatic detection and classification of various categories of microcalcification. The traditional image processing approach for automatic microcalcification detection consists of preprocessing, segmentation, feature extraction, and classification steps. These steps are indirectly included in various layers of a deep learning model trained by a large portion of relevant data cases. Segmentation is an important step preceding classification since feature extraction is highly affected by the local area of operation, and the accuracy of the microcalcification detection depends on the accuracy of its segmentation. In this medical field, Segmentation and feature extraction become difficult due to variations in shape, directionality, contrast, and size of images and microcalcifications, which are hidden in dense breast tissue. Conventional segmentation schemes are based on stochastic modeling of image characteristics, especially focusing on the region of interest of the mammographic image of utmost interest to the medical expert. In contrast, deep learning avoids several inefficiencies of model-based classification by exploiting knowledge and wishful thinking oriented from the data themselves. Nevertheless, the classification task in deep learning does not specifically utilize the benefits of segmentation.

In this study, we explored whether classification accuracy can be significantly enhanced by imposing the segmentation step in preprocessing and improving the feature extraction stage, using deep learning models. More specifically, our methodology operates on local windows as the regions of interest, in the form of a moving window approach scanning the entire mammographic image. Each time, we focused on a specific region of interest and stochastic modeling of its anatomic structure. Subsequently, we exploited a pretrained network in the form of transfer learning, to serve as an effective feature extractor based on a large set of image structures. More specifically, we exploited the advantages of efficiency vs. complexity of InceptionResNetV2 adapted to the framework of our application. Following the feature extraction on microcalcification images from this pretrained network, we proceeded with the training of the classification layer of a deep learning model, to assess the performance of the entire system. In order to further enhance the performance of the deep neural network, we exploited a number of optimizers for parameter tuning in the classification layer. We tested the proposed deep learning model on a widely used mammogram dataset (as it is described in Section 2.1) and compared its performance with classical machine learning schemes that perform tedious, texture-based feature extraction, as well as other previously tested deep learning schemes that operate in a black-box form on the data. Our model outperforms such approaches, due to (1) the appropriate restriction of the data domain of CNN operation, extracting detailed structural features within local windows, and (2) the appropriate design of filter concatenation in the InceptionResnetV2 model and the efficient adaptation of the output layers through optimizers.

This paper is structured into the following sections: Material and methods are described and Experimental analysis of detection and classification of microcalcification is discussed in Section 2, followed by results and discussion in Section 3. Finally, Section 4 concludes the findings of this study.

## 2. Materials and Methods

### 2.1. Dataset Description

The performance of the proposed deep learning framework for the classification of microcalcification was evaluated using the open, curated breast imaging subset of the DDSM mammogram dataset (CBIS–DDSM) [15]. This is a standard version of digital database screening mammography (DDSM), which consists of benign, benign without a callback, and malignant cases. This study considered the detection and classification of microcalcification into three categories—namely, benign, malignant, and benign without a callback. Benign microcalcification represents the suspicious cancerous region that may require further investigation by another modality such as ultrasound or biopsy. The case of benign without a callback indicates that the region may be suspicious and should be monitored, but it does not require further investigation at the moment. Malignant microcalcification indicates the suspicious regions of cancer, which requires proper evaluation, recommendation, and treatment by the physician. This research mainly focused on the classification of three categories of microcalcification. Cropped region of interest (ROI) images including various categories of microcalcification were used to assess the performance of our proposed classification scheme. These ROIs reflected all types of abnormalities, as defined by medical experts, and the local properties of regions that be involved in our moving window operation of the proposed detection scheme. The dataset was split into 80% for training and hyperparameter tuning and 20% for stratified testing sets based on Breast Imaging Reporting and Data System (BIRDS) category. In total, 1547 and 326 ROI images of the CBIS–DDSM dataset were used for training and validation, respectively. Table 1 shows the number of ROI images from all cases, consisting of benign without a callback, benign, and malignant types of microcalcification.

### 2.2. Image Preprocessing

Sample images of the DDSM dataset of each category are shown in Figure 1, consisting of squared ROI images of different sizes. These images were converted to a standard size of 299 × 299, using inter-cubic interpolation algorithms, as the convolutional neural network (CNN) structure operated on images of the same size.

DDSM mammography data were single-channel image sets, different from natural color images applied in the pretrained models. However, basic image features in terms of edges, shapes, and other high-level features can still be extracted using pretrained convolutional neural network-based models. In this respect, we note that the sample DDSM image was a single-channel image, whereas the input to the InceptionResNetV2 model required a three-channel image. Thus, sample DDSM images were converted to three channels by copying the pixel value of single-channel images to the other two channels.

### 2.3. Transfer Learning

A deep neural network with several hidden layers is expensive to train. Complex models require multiple machines with expensive GPUs and still may take weeks to train them. Transfer learning is a popular approach in deep learning in which pretrained models are used as the starting point of the neural network model. The main field application is computer vision and natural language processing. There are different variants of pretrained networks available, each with its own architecture, speed, size, advantages, and disadvantages. Some of the most common available models include VGGNET, RESNET, or INCEPTION [16,17,18].

In this study, the InceptionResNetV2 model was utilized, pretrained with the ImageNet dataset. Mammograms as images are different from natural images due to the number of channels, as well as the embedded complexity of their structure. Thus, the pretrained InceptionResNetV2 was used only to obtain the feature maps for subsequent classification purposes. The actual training of the classification scheme of microcalcification into three categories was performed on the CBIS–DDSM dataset.

The structure of the convolutional neural network (CNN) architecture adapted in our model is shown in Figure 2. The image was inputted to the InceptionResnetV2 model after resizing the cropped ROIs of the original image into 299 × 299 × 1.

The pretrained InceptionResNetV2 model on the ImageNet dataset is shown in Figure 3. The size of input images of the InceptionResNetV2 model is 299 × 299 × 3, and its full architecture is shown in the middle panel of Figure 3. The initial STEM operator decoupled the input into many channels of smaller size, resulting in a cube of 35 × 35 × 384, as shown in the left panel of Figure 3. This was followed by two InceptionResNet blocks, each followed by its specific STEM reduction operator with appropriately designed filter concatenations, as illustrated in the right panel of Figure 3. We used different filter concatenations to gradually reduce the size of the grid from 35 × 35 to 17 × 17 and then to an 8 × 8 grid. A third InceptionResNet block with average pooling on the output dimensions provided the flattened output, which was trained for classification with SoftMax.

In our proposed architecture (Figure 2), the flattened output of this pretrained model was employed as input to a fully connected layer with 128 neurons, followed by an output layer. The ReLU and SoftMax activation functions were utilized in the fully connected layer and the output layer, respectively. In particular, the RELU activation restricted negative values and forced many hidden layers to zero, thus providing a sparse representation. Furthermore, ReLU acted as a linear activation function, facilitating the efficient optimization of the neural network. On the other hand, the SoftMax activation in the output layer of classification derived a multinomial probability distribution and was adjusted for three classes—namely, benign, benign without a callback, and malignant.

The proposed CNN architecture was tested with different optimizers for the classification of microcalcifications. Different multiclass classification loss functions were tested in general, including multiclass cross-entropy and Kullback–Leibler (KL) divergence loss [19,20]. This study utilized Kullback–Leibler divergence as a loss function, which calculates the information loss if the predicted probability distribution is used to approximate the desired target.

### 2.4. Experimental Analysis

The convolutional neural network was implemented using the Tensor Flow machine learning technique. We used transfer learning with the pretrained InceptionResNetV2 model for feature extraction and further train the classification layer. The simulation was performed on a 3.2 GHz processor with 32 GB Memory, using Python. The experiment was tested and validated on the DDSM dataset. The dataset was portioned into three sets for training, testing, and validation. In total, 1547, 326, and 200 ROI images of the CBIS–DDSM dataset were used for training, testing, and validation, respectively. In order to tune the network parameters in an automated manner without the intervention of an operator, various optimizers were applied in this study—namely, ADAM, ADAGrad, ADADelta, and RMSProp [21,22]. Adam is a replacement optimization algorithm for stochastic gradient descent used in trained deep learning models. Adam builds on the properties of ADAGrad algorithms to provide an optimization algorithm that can handle sparse gradients in noisy-problem environments. Faster convergence of deep learning models can be obtained using momentum and adaptive learning rates. The complete list of architectures is presented in Table 2. The learning rate of various optimizers was tuned from 0.01 to 0.00001. Finally, the learning rate set was taken as 0.0001. The exponential decay rate for the 1st and 2nd moments for the ADAM optimizer was set to default values of 0.9 and 0.999, respectively.

### 2.5. Performance Evaluation

A total of 1547 training and 326 testing ROI images were used in this research. The dataset was split into training and validation set based on the BIRDS category. Each of the positive images contained at least one biopsy-proven malignant tumor. All examples in the dataset were taken in either CC or MLO view or both. Cranial–caudal (CC) is a view from above, while mediolateral–oblique (MLO) is an oblique or angled view. Rectangular regions of interest from CC and MLO views of a mammogram were extracted and then converted into square ROI by stretching the smaller side. The performance of each CNN prediction model was evaluated by computing the area under the receiver-operating characteristic curve (ROC), sensitivity, and specificity, which are defined as follows [19,20]:(1)$$\mathrm{True}\;\mathrm{Positive}\;\mathrm{Rate}\;=\frac{\mathrm{True}\;\mathrm{Positives}\;}{\left(\mathrm{True}\;\mathrm{Positives}\;+\;\mathrm{False}\;\mathrm{Negatives}\right)}$$
(2)$$\mathrm{False}\;\mathrm{Positive}\;\mathrm{Rate}\;=\frac{\mathrm{False}\;\mathrm{Positives}\;}{\left(\mathrm{False}\;\mathrm{Positives}\;+\;\mathrm{True}\;\mathrm{Negatives}\right)}$$
(3)$$\mathrm{Sensitivity}=\frac{\mathrm{True}\;\mathrm{Positives}\;}{\left(\mathrm{True}\;\mathrm{Positives}\;+\;\mathrm{False}\;\mathrm{Negatives}\right)}$$
(4)$$\mathrm{Specificity}\;=\frac{\mathrm{True}\;\mathrm{Positives}\;}{\left(\mathrm{True}\;\mathrm{Negatives}\;+\;\mathrm{False}\;\mathrm{Positives}\right)}$$
(5)$$\mathrm{Accuracy}=\frac{\mathrm{True}\;\mathrm{Positives}+\mathrm{True}\;\mathrm{Positive}}{\left(\mathrm{True}\;\mathrm{Positive}+\mathrm{True}\;\mathrm{Negatives}\;+\;\mathrm{False}\;\mathrm{Positives}+\mathrm{False}\;\mathrm{Negatives}\right)}$$

ROC curves were plotted as true-positive versus false-positive rates at various thresholds. The area under the curve (AUC) was used for a better understanding of ROC.

## 3. Result and Discussion

The proposed CNN was tested with four different optimizers—namely, ADAM, ADAGrad, ADADelta, and RMSProp. The parameters of the CNN were fine-tuned with each of these optimizers, to attain high classification results. Loss, accuracy, and sensitivity were plotted for each optimizer, with 20 epochs of training and validation, as described in the following subsections. The smaller batch size was used to achieve better stability and generalization of the model. Therefore, in this study, the batch size was taken as 32 for the experiment.

### 3.1. Implementation of InceptionResNetV2 with ADAM Optimizer

Various parameters of the ADAM optimizer were tuned to obtain the highest classification score. The learning rate was set to 0.0001, whereas the exponential decay rates for first- and second-moment estimates were taken as 0.9 and 0.999, respectively. The value of epsilon, which is a small constant for numerical stability, was set to the default value of 1 × 10^–7^. The result of Inceptionresnetv2 with ADAM optimizer is shown in Figure 4, presenting the evaluation measures in training and validation stages.

### 3.2. Implementation of InceptionResNetV2 with ADAGrad Optimizer

The learning rate of ADAGrad is a variable parameter, depending on how frequently it is updated [21]. This learning rate and initial accumulator value of ADAGrad were tuned as 0.0001 and 0.1, respectively. Epsilon value was set as a default value of 1 × 10^–7^. The result of InceptionResNetv2 with ADAGrad optimizer is shown in Figure 5.

### 3.3. Implementation of InceptionResNetV2 with ADADelta Optimizer

ADADelta is a more robust extension of ADAGrad that performs learning based on a moving window of gradient updates, instead of accumulating all previous gradients. By this method, ADADelta continues adjusting even after it updates the learning rate. The learning and decay rates were initialized as 0.0001 and 0.95, respectively. Epsilon was taken with a default value of 1 × 10^–7^. The result of InceptionResNetv2 with ADADelta optimizer is shown in Figure 6.

### 3.4. Implementation of InceptionResNetV2 with RMSProp Optimizer

RMSProp optimizer works by maintaining a moving (discounted) average of the square of gradients and dividing the gradient by the root of this average [22]. The learning rate for RMSProp was initialized as 0.0001. The dividing factor (or rho) and the momentum parameter were initialized as default values of 0.9 and 0.0, respectively. Epsilon value was initialized as a default value of 1 × 10^–7^. The result of InceptionResNetv2 with RMSProp optimizer is shown in Figure 7.

The test results of the proposed CNN models with different optimizers are summarized in Table 3. It is revealed that InceptionResNetV2 is an effective transfer learning model for microcalcification classification. By analyzing the results, it can be concluded that this pretrained model is more sensitive in detecting key elements such as edges and shapes within a mammogram image. In this study, the inceptionResNetV2 pretrained model with Kullback–Leibler divergence loss function and combination of various optimizers proved an effective model for multiclass classification of microcalcification [23,24].

The proposed CNN model performs efficiently irrespective of the optimizers’ type. The highest training accuracy of 98% is achieved with 0.0212 training loss. The highest validation accuracy, AUC, sensitivity, and specificity values were 94%, 96%, 97%, and 80%, respectively. Traditional machine learning approaches were also tested, to compare with the results of the deep learning approach. In total, 14 second-order gray-level co-occurrence matrix (GLCM) features were extracted, to generate texture attributes [22]. The feature vector set was given as input to SVM and k-NN classifiers trained to classify the various types of microcalcification. The classification performance was measured by 10-fold cross-validation. The classification results of SVM with RBF kernel function and k-NN classifier are shown in Table 4.

The outcomes of the present study were compared with the results of existing approaches, in order to illustrate the benefits of adopting segmentation in preprocessing and efficient optimizers in the classification, as illustrated in Table 5. Ribli et al. achieved the highest AUC value of 0.95 with a faster R-CNN deep learning model, to detect and classify breast lesions using mammograms on IN Breast dataset [11]. Arevalo et al. utilized CNN for mass classification, achieving the highest AUC value of 0.82 on 736 film mammograms [12]. Dhungel et al. achieved 98% sensitivity with a deep learning model for the detection of masses present in the mammograms [13]. Becker et al. achieved 81% accuracy with a deep artificial neural network model for breast cancer detection on digital mammograms [14]. As demonstrated by Table 5, the proposed deep learning model performs better than counterpart existing models.

In the present study, we utilized InceptionResNetV2 with four different optimizers, for the classification of an openly available mammogram dataset including three classes. A future step of this research could potentially be the examination of more classes, i.e., normal mammogram images, as well as other types of abnormalities.

## 4. Conclusions

An effective deep-learning-based approach was proposed and utilized to classify the three important categories of breast tissue cancer, as benign without a callback, benign, and malignant microcalcification, using mammograms. Manual feature extraction of abnormal ROI was avoided using deep learning approaches. In our case, the pretrained InceptionResNetV2 model was used for automatic feature extraction in local regions of interest. Various optimizers including ADAM, ADAGrad, ADADelta, and RMSProp were also used to fine-tune the parameters of the InceptionResNetV2 pretrained model. The highest training rate of 98% and validation accuracy of 94% were achieved with ADADelta optimizer with a learning rate of 0.001. During training, the highest training rate of 98% and validation accuracy of 94% were achieved with ADADelta optimizer with a learning rate of 0.001. In the testing phase, 97% sensitivity, 80% specificity, 94% accuracy, and 96% AUC were achieved with our proposed classification scheme, outperforming the performance of previously used deep learning approaches and classical machine learning schemes.

## Figures and Tables

**Figure 1 ijerph-19-02159-f001:**
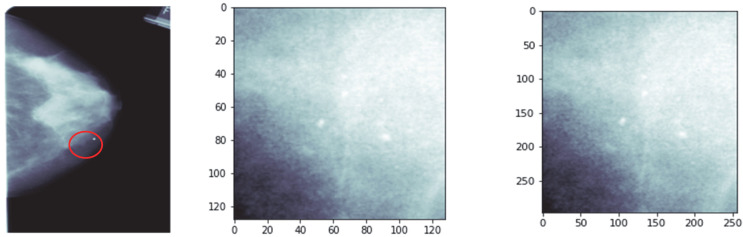
(**a**) Original mammogram images taken from DDSM dataset; (**b**) cropped ROI images of original images; (**c**) resized ROI images of size 299 × 299.

**Figure 2 ijerph-19-02159-f002:**
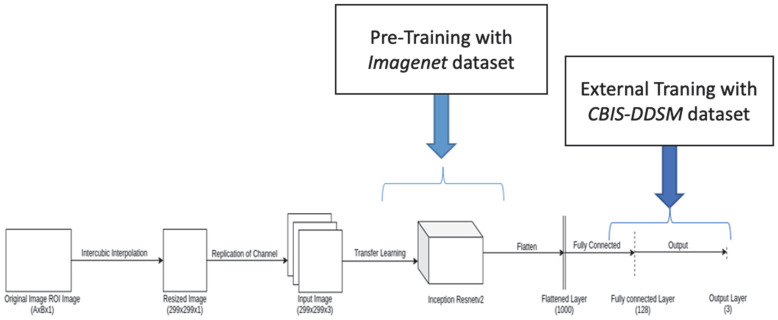
Proposed convolutional neural network architecture.

**Figure 3 ijerph-19-02159-f003:**
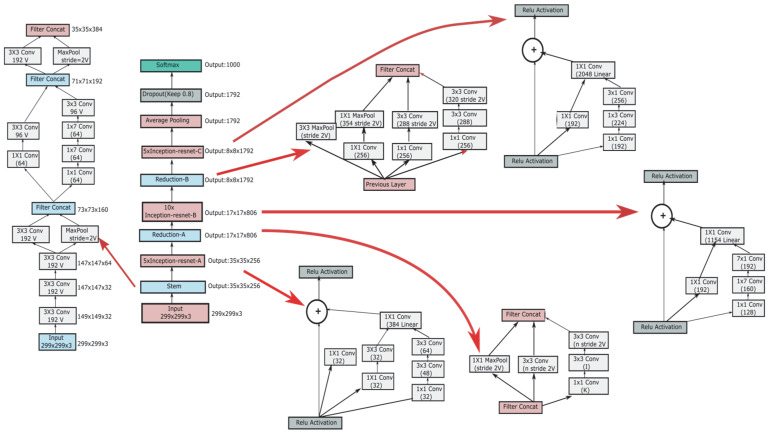
Architecture of CNN model.

**Figure 4 ijerph-19-02159-f004:**
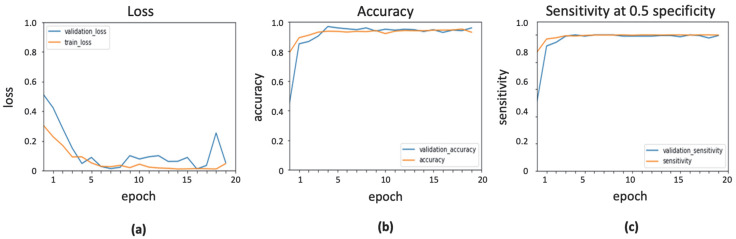
Output obtained using ADAM optimizer: (**a**) loss function; (**b**) accuracy; (**c**) sensitivity.

**Figure 5 ijerph-19-02159-f005:**
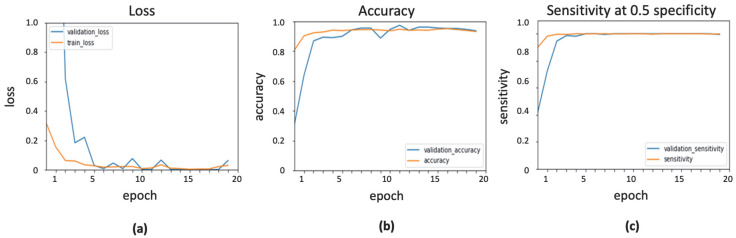
Output obtained using ADAGrad optimizer: (**a**) loss function; (**b**) accuracy; (**c**) sensitivity.

**Figure 6 ijerph-19-02159-f006:**
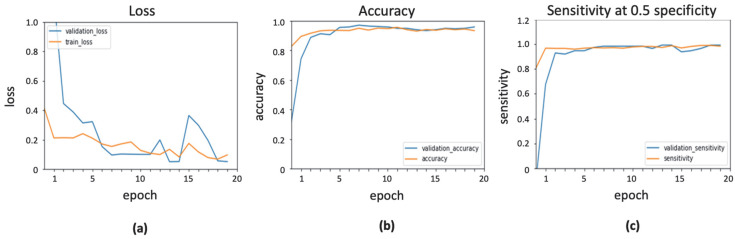
Output obtained using ADADelta optimizer: (**a**) loss function; (**b**) accuracy; (**c**) sensitivity.

**Figure 7 ijerph-19-02159-f007:**
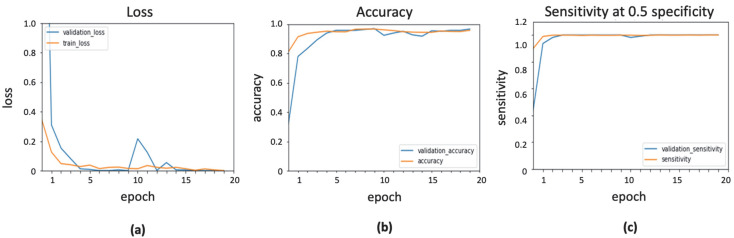
Output obtained using RMSProp optimizer: (**a**) loss function; (**b**) accuracy; (**c**) sensitivity.

**Table 1 ijerph-19-02159-t001:** The number of images used in each class.

	Training	Testing
Benign_without_callback	474	99
Benign	528	133
Malignant	545	94

**Table 2 ijerph-19-02159-t002:** Architecture of transfer learning model (InceptionResNetV2) with various optimizers.

Optimizer	Input Shape	Fully Connected Neurons	Fully Connected Activation Function	Output	Output Activation Function
Adam	299 × 299 × 3	128	Relu	3	Softmax
AdaGrad					
AdaDelta					
RMSProp					

**Table 3 ijerph-19-02159-t003:** Validation results of various models with a learning rate of 0.0001 and a batch size of 32, with model names and loss functions.

Model	Loss Function	Optimizer	Training Loss	Training Accuracy
Inception ResNetV2	Kullback_Leibler_ Divergence	ADAM	0.1134	0.9813
Inception ResNetV2	Kullback_Leibler_ Divergence	ADAGrad	0.0212	0.9813
Inception ResNetV2	Kullback_Leibler_ Divergence	ADADelta	0.1293	0.9816
Inception ResNetV2	Kullback_Leibler_ Divergence	RMSProp	0.1193	0.9810

**Table 4 ijerph-19-02159-t004:** Validation result of various models with a learning rate of 0.0001 and a batch size of 32 with InceptionResNetV2 Model and Kullback_Leibler_ Divergence loss function.

Model	Loss Function	Optimizer	Loss	Accuracy	AUC	Sensitivity at Specificity 0.8
Inception ResNetV2	Kullback_Leibler_Divergence	ADAM	0.21	0.93	0.95	0.96
Inception ResNetV2	Kullback_Leibler_Divergence	ADAGrad	0.67	0.93	0.93	0.93
Inception ResNetV2	Kullback_Leibler_Divergence	ADADelta	0.28	0.94	0.96	0.97
Inception ResNetV2	Kullback_Leibler_Divergence	RMSProp	0.32	0.92	0.95	0.95
SVM(RBF Kernel function)	-	-	-	0.91	0.90	91
k-NN				0.89	0.88	0.89

**Table 5 ijerph-19-02159-t005:** Comparison of the proposed model with existing techniques.

Article	Model	Accuracy (%)	AUC	Sensitivity (%)
Ribli et al. [11]	faster R-CNN	0.92	0.95	96
Arevalo et al. [12]	CNN	0.90	0.82	85
Dhungel et al. [13]	CNN	0.92	0.93	98
Becker et al. [14]	CNN	81	0.89	87
Proposed DL model with ADAM	Inception ResNetV2	0.93	0.95	0.96
Proposed work DL model with ADAGrad	Inception ResNetV2	0.93	0.93	0.93
Proposed work DL model with ADADelta	Inception ResNetV2	0.94	0.96	0.97
Proposed work DL model with RMSProp	Inception ResNetV2	0.92	0.95	0.95

## Data Availability

The data presented in this study are openly available in https://doi.org/10.1038/sdata.2017.177 (accessed on 4 January 2022) at reference No. [15].

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
