# Peer review of "Deep Learning Capabilities for the Categorization of Microcalcification"

_ijerph, 2022, doi:10.3390/ijerph19042159_

Round 1

Reviewer 1 Report

This manuscript proposed a convolution neural networks for the automatic detection and classification of all three categories of micro-calcification, which outperforms the ability of previously used deep learning approaches and classical machine learning schemes.
The method authors proposed seems work but have many questions on this manuscript.

1. The number of images shown in Table 1 is not sufficient for a large CNN network. The authors should show the details of train process.

2. The details of Figure2 cannot be clear read clearly.

3. Why select the architecture of CNN model shown in Figure 3.

4. The experimental results do not seem clear enough. The author needs to describe the experiment more clearly.

In general, this manuscript needs some improvement to meet the publishing requirements.

Author Response

We thank the reviewer for the valuable comments/suggestions which can potentially increase the quality of our work. All text changes are highlighted in red in the revised manuscript. Please find in the following all responses to the reviews.

  1. The number of images shown in Table 1 is not sufficient for a large CNN network. The authors should show the details of train process.

Ans.  In total, we employed 1873 images for the experiments having micro-calcification of three categories. Out of the total images, a total of 1547 images have been taken as training data, and a total of 326 images is taken as test data. The pre-trained CNN model is used for training and testing. Our CNN model is based on the transfer learning approach so that the training does not pose a heavy burden on the number of training samples. In this deep learning approach pre-trained models are used as the starting point of the neural network model. The most common available models include VGGNET, RESNET, or INCEPTION which has been used to obtain the feature maps for subsequent classification of input images of the training set.

  1. The details of Figure2 cannot be clear read clearly.

Ans. We re-created Fig. 2 and explained it clearly.

  1. Why select the architecture of CNN model shown in Figure 3?

Ans.  The CNN is fully connected network in which all neurons are connected with each other. This model has been used for image classification and recognition with high accuracy. The detailed architecture of the CNN model is presented in Fig. 3. This figure helps the reader to understand the implementation of the proposed model. The details of various parameters and optimizers taken for the CNN model in the experiment are presented in Fig. 3, which shows the details of the execution of the CNN model. The ReLU and Softmax activation functions are utilized in the fully connected layer and the output layer.

  1. The experimental results do not seem clear enough. The author needs to describe the experiment more clearly.

Ans. The present experiment is tested and validated on the DDSM dataset. The dataset is partitioned into three set for training, testing, and validation. In total, 1547, 326, and 200 ROI images of the CBIS-DDSM dataset are used for training, testing, and validation, respectively. In this work, the InceptionResNetV2 model is utilized within a transfer-learning environment, as pre-trained with the Imagenet dataset. The mammogram images of the DDSM dataset are different from natural images used to train the original InceptionResNetV2 model, due not only to the number of channels but also to the embedded complexity of their structure. Thus, the pre-trained InceptionResNetV2 is used only to obtain the feature maps for subsequent classification purposes. The actual training of the classification scheme is performed on the CBIS-DDSM dataset. We modified the corresponding text in section 2.4 (see text in red).

Reviewer 2 Report

Authors propose a deep learning-based approach to classify three important categories of breast tissue cancer and provide analysis of the network for different optimizers. Although the results look convincing, I have a few concerns that needs to be addressed:

1) Contribution of this paper are not clear. Please talk about your proposed scheme in the abstract and also briefly say how it is unique from existing works. Later in the end of introduction, add a few lines or bullets stating the novelty of this work.

2) Quality of figures need to be improved. In few figures, it is hard to read the text clearly. Also, please add reference to InceptionResNetV2 in Figure 3 and mention the source for Figure 1.

3) The results table looks incomplete. No accuracies for existing methods are mentioned. Authors should make efforts to implement the code of existing works and report all metrics. 

4) Please review "Computer Vision-Based Microcalcification Detection in Digital Mammograms Using Fully Connected Depthwise Separable
Convolutional Neural Network" for the similar existing works. I believe there are a plenty of works that use different deep learning approach for micro-classification. 

Author Response

We thank the reviewer for the important comments/suggestions which are very helpful for our work. Please find in the following all responses to the reviews. All text changes are highlighted in red in the revised manuscript. Please find in the following all responses to the reviews.

  • Contribution of this paper are not clear. Please talk about your proposed scheme in the abstract and also briefly say how it is unique from existing works. Later in the end of introduction, add a few lines or bullets stating the novelty of this work.

Ans.  The contribution and novelty of the proposed work are mentioned in the manuscript on page number 3, paragraph number 2. The proposed scheme is also presented on page number 3, paragraph number 3.  More specifically, diverting from existing works, we explore if classification accuracy can be significantly enhanced by imposing the segmentation step in preprocessing and improving the feature extraction stage using deep learning models. Our proposed methodology operates on local windows as the regions of interest from the mammographic image and segments it using stochastic modeling of its anatomic structure. Subsequently, we exploit a pre-trained network in the form of transfer learning, as an effective feature extractor based on a large set of local mammogram sub-structures.

  • Quality of  figures need to be improved. In few figures, it is hard to read the text clearly. Also, please add reference to InceptionResNetV2 in Figure 3 and mention the source for Figure 1.

Ans.  Fig. 2 and Fig. 3 have been redrawn. The source of Figure 1 has been added in the caption of Figure.

  • The results table looks incomplete. No accuracies for existing methods are mentioned. Authors should make efforts to implement the code of existing works and report all metrics.

Ans. The performance of the proposed model is measured in terms of Accuracy, Sensitivity, Specificity, AUC (Area Under Curve), and loss, according to the reviewer’s comment. The values of all five parameters obtained with training and testing data set with different optimizers are presented in Table 3 and Table 4, respectively.  

Reviewer 3 Report

This is an interesting work, however some points require clarification:

  1.  In Fig 1 c, please verify if the figure is of size 299 x 299 since the x axis seems to have limits close to 120.
  2. Why was no validation set used for evaluating the performance of the dataset?
  3. Fig 2 seems to be cropped incorrectly.
  4. Please include the batch size in section 3 description.
  5. In line 159, what are the actual number of cancer cases used for training, testing and validation? To prevent overfitting, it would be best if the same proportion of true positives are used for each category.
  6. In table 5, do all the models use the same CBIS-DDSM dataset? If not consider removing them, since testing was not done on other independent datasets. 
  7. In table 5, what are the estimated metrics (eg. AUC, specificity, sensitivity, accuracy) for the considered models?
  8. Please include the github repo for the proposed model.
  9. Line 232 mentions multi-class cross-entropy loss function was used in addition to KL loss. Please provide the results of the same.

Author Response

We thank the reviewer for the detailed comments/suggestions that make our work better. All text changes are highlighted in red in the revised manuscript. Please find in the following all responses to the reviews. Please find in the following all responses to the reviews.

  1. In Fig 1 c, please verify if the figure is of size 299 x 299 since the x axis seems to have limits close to 120.

Ans.  Fig. 1(b) is a size of 120×120 which is the original cropped image. The resizing of the original cropped image has been done as 299 x 299 which is shown in Fig. 1 in the updated manuscript.

  1. Why was no validation set used for evaluating the performance of the dataset?

Ans. The model has been validated with 200 ROI of mammogram images and validation results are shown in Table 4.

  1. Fig 2 seems to be cropped incorrectly.

Ans.  Fig. 2 is redrawn and corrected in the updated manuscript.

  1. Please include the batch size in section 3 description.

Ans.  Batch size is mentioned in Section 3 as:

The smaller batch size is used to get better stability and generalization of the model. So, the batch size is taken as 32 in this work for the experiment.

  1. In line 159, what are the actual number of cancer cases used for training testing and validation? To prevent overfitting, it would be best if the same proportion of true positives are used for each category.

Ans. In this paper classification of three categories of micro-calcification has been performed. The number of all three classes of micro-calcification used for training and testing is given in table 1.

Table 1. The number of images used in each class.

Training

Testing

BENIGN_WITHOUT_CALLBACK

474

99

BENIGN

528

133

MALIGNANT

545

94

  1. In table 5, do all the models use the same CBIS-DDSM dataset? If not consider removing them, since testing was not done on other independent datasets.

Ans.   In Table 5, all models have been implemented with the CBIR-DDSM data set and obtained the results.

  1. In table 5, what are the estimated metrics (eg. AUC, specificity, sensitivity, accuracy) for the considered models?

Ans.  The mathematical formula and their importance of AUC, specificity, sensitivity, and accuracy are given in the manuscript in Eq. 1 to Eq 5. The values obtained for the metrics are listed in table 5.

  1. Please include the github repo for the proposed model.

Ans. The present study is part of a large ongoing project and thus not possible to make it available at the moment. However, we intend to make the entire repository available after the end of the project.  

  1. Line 232 mentions multi-class cross-entropy loss function was used in addition to KL loss. Please provide the results of the same.

Ans Both Multi-Class Cross-Entropy and KL Loss represent the Multi-Class Classification Loss Functions. We can use any one of the loss functions. In this work, the KL loss function has been used to reduce the error.

Round 2

Reviewer 1 Report

This revised version manuscript seems better than before.  

Please check the english grammar and detailed data, and make minor update before publishing.

Reviewer 2 Report

The authors did a good job in revising the manuscript and addressing the comments from previous review. 

Reviewer 3 Report

The changes made are acceptable.